# A Molecular Orchestration of Plant Translation under Abiotic Stress

**DOI:** 10.3390/cells12202445

**Published:** 2023-10-13

**Authors:** Aleksandra V. Suhorukova, Denis S. Sobolev, Irina G. Milovskaya, Vitaliy S. Fadeev, Irina V. Goldenkova-Pavlova, Alexander A. Tyurin

**Affiliations:** Laboratory of Functional Genomics, Timiryazev Institute of Plant Physiology, Russian Academy of Sciences, 127276 Moscow, Russia; sualsha@yandex.ru (A.V.S.); denissoboleww@gmail.com (D.S.S.); irina.20152016@mail.ru (I.G.M.); lis_vit@rambler.ru (V.S.F.); irengold58@gmail.com (I.V.G.-P.)

**Keywords:** abioitic stress, plant translation, IRES, 5′-UTR, uORF, codon bias, G-qudruplex

## Abstract

The complexities of translational strategies make this stage of implementing genetic information one of the most challenging to comprehend and, simultaneously, perhaps the most engaging. It is evident that this diverse range of strategies results not only from a long evolutionary history, but is also of paramount importance for refining gene expression and metabolic modulation. This notion is particularly accurate for organisms that predominantly exhibit biochemical and physiological reactions with a lack of behavioural ones. Plants are a group of organisms that exhibit such features. Addressing unfavourable environmental conditions plays a pivotal role in plant physiology. This is particularly evident with the changing conditions of global warming and the irrevocable loss or depletion of natural ecosystems. In conceptual terms, the plant response to abiotic stress comprises a set of elaborate and intricate strategies. This is influenced by a range of abiotic factors that cause stressful conditions, and molecular genetic mechanisms that fine-tune metabolic pathways allowing the plant organism to overcome non-standard and non-optimal conditions. This review aims to focus on the current state of the art in the field of translational regulation in plants under abiotic stress conditions. Different regulatory elements and patterns are being assessed chronologically. We deem it important to focus on significant high-performance techniques for studying the genetic information dynamics during the translation phase.

## 1. Introduction

The study of the response of plants to stress remains an urgent task in plant physiology. The fundamental component of this direction is complemented by the most important applied aspects: the maintenance of and increase in crop yields, the production of genetically modified plants, the fight against the consequences of environmental disasters, etc. The purpose of this review is to consider the main aspects of the study of translation in plants under stress conditions, namely modulation of activity, changes in the profile of translated genes, patterns and mechanisms of regulation, as well as the methods used in this field of research, both classical and those developed specifically for this task. However, it is impossible to consider translation in isolation from all the other stages in the implementation of genetic information. Stress affects all aspects of plant life, from the structure and functional activity of the genome to the metabolome and physiological responses. The process of metabolism in all living organisms is highly regulated due to evolutionary processes. In the present review, we focus on the sequential consideration of the process of plant response to the action of stress factors.

A plant organism’s recognition of stressful conditions and its response to them are formed due to a developed sensory system; Figure 1. This set of sensors includes, first of all, cell membrane sensor proteins. Such proteins respond to changes in temperature, small signal molecules, changes in membrane fluidity, etc. Curiously, the sensory system of a plant cell is not limited to external sensors. For example, in recent years, researchers have expressed significant interest in sensors that express an unfolded protein response. Proteins that perform this function are located in the endoplasmic reticulum and recognise misfolded or not completely folded protein molecules, signalling the development of stressful conditions inside the cell. In particular, the tomato model [1,2] and, of course, Arabidopsis [3,4] demonstrate the active use of this molecular mechanism in plants.

The initial mRNA population that enters the translation pathway does not come from nowhere. A set of molecular instructions is created during transcription. But this is not the beginning. In addition to the nucleotide context (promoters and enhancers, etc.), various epigenetic processes are also involved in the permanent storage of genetic information in the genome. Epigenetic processes alter the availability of genes for translation. Abiotic stress has a broad impact on the process of expression regulation and, of course, affects the epigenetic aspect. Since a detailed consideration of epigenetic regulation in stressed plants is beyond the scope of this review, we briefly review current work dealing only with chromatin modulation and DNA methylation. The genome is usually not directly accessible to the effects of stress factors. Using *Arabidopsis thaliana* as an example, it can be seen that stress factors trigger different types of epigenetic responses through signalling pathways (e.g., the jasmonate pathway). For example, it can be seen that different types of abiotic stress change the number of open chromatin regions (OCR) differently. Drought and high temperature increase the number of similar genomic regions by 26 and 16 per cent, respectively, compared to the control, whereas low-temperature stress reduces the representation of these regions by 18 per cent [5].

Using rice (*Oryza sativa*) as an example, global changes in chromatin structure, including changes in the size of topologically associating domains, loss of short-range interactions and A/B compartment transition, have been demonstrated in response to heat stress [6]. Heat stress also induces activation of transposable elements (TEs), with the degree of activation negatively correlated with the degree of chromatin interaction [7]. Methods for artificial epigenetic modification of the plant genome are of particular interest. This approach opens up additional possibilities for regulating expression, including under conditions of abiotic stress. A major contribution to the development of these methods has been made by modification of the genome editing technology (CRISPR:Cas9) [8]. To further explore the epigenetic regulation of gene expression in plants under stress, two comprehensive reviews are recommended: [9,10].

The stages of post-transcriptional modification, processing and splicing, also play an important role in plant acclimation to stress. It is known, for example, that under the influence of changing temperature, the mechanism of alternative splicing can be activated, which not only generates the temperature isoforms of various enzymes, but also modifies the 5′-untranslated region (5′-UTR) [11]. Such changes in the composition of the 5′-UTR can affect its structure in a number of ways, altering the permeability of this region of the mRNA to ribosomes.

Billions of years of evolution have formed a very complicated and at the same time finely tuned mechanism of translation; Figure 2. Translation as the final step in the transfer of genetic information is described in detail both in terms of chemical kinetics and as a complex multi-component process [12,13]. Translation is a complex, multi-step process. The efficiency of gene expression at the translation stage depends on a number of key players: translation factors, uORFs (upstream open reading frames), IRES (internal ribosome entry site), riboswitches (including RNA thermometers), codon composition, etc. Translation also has its own stages, usually distinguished by initiation, elongation and termination. The key stage is usually considered to be translation initiation. The two main pathways by which translation initiation can occur are CAP-dependent and CAP-independent translation initiation; Figure 3. The CAP-dependent mechanism involves the recognition of 5′-CAP (7-methylguanosine) 43S by the pre-initiation complex. The 43S complex in turn consists of translation initiation factors: eIF3, eIF4, eIF2 (more specifically, eIF2:GTP), Met-tRNA and, of course, the small subunit (40S) of the ribosome. The eIF2B factor is also required. Translation elongation begins when the entire complex completes its scan and is positioned at the start codon (AUG). Next, the large subunit of the ribosome attaches to the preinitiation complex and the translation initiation factors are released. Cap-independent translation is best studied using IRES. In this case, the ribosome does not scan the mRNA for a start codon, but is recruited directly to a specific entry site (IRES) with the participation of IRES trans-acting factors. Typically, two key factors are required for elongation: eEF1 (which provides the necessary tRNAs) and eEF2 (which translocates the mRNA relative to the ribosome). The termination of translation involves the release of stop codon recognition factors. For details of the regulation of translation in plants under normal conditions, see [13].

Historically, the study of the influence of abiotic factors on the plant organism and the response of plants to these factors began with classical physiological articles. A considerable amount of data has also been obtained using biochemical approaches. However, the most active idea of responding to and overcoming the effects of abiotic environmental factors began to advance with the advent and development of molecular genetic methods and hybrid approaches that also incorporate classical biochemical techniques. Along with this important methodological branch in this field, the analysis of mutants that have lost the genes responsible for plant acclimation under stressful conditions has been and still is pending.

In the present review, we attempt to update the most recent ideas on the regulation of translation in plants under conditions of abiotic stress. Taking into account the fact that many regulatory elements can manifest themselves at all the major stages of translation, we decided to consider each type of regulatory code separately, without reference to the stages of translation.

## 2. Regulation of Plant Translation under Stress Conditions

### 2.1. Translation Factors Are a Necessary Part of Translation

The best studied element of translation control is initiation. It is currently thought that initiation has the greatest influence on regulation, including during various stresses. However, it is initiation that has distinct components for different kingdoms, in contrast to elongation and termination, which are the most conservative. In this section, translation initiation factors are discussed, but it should be noted that within the plant kingdom, there is great diversity in the translation apparatus, and most current ideas are based on data obtained from *A. thaliana*. According to modern concepts, the number of translation initiation factors for eukaryotes exceeds fifteen. In this section, we consider only those factors that have the most direct effect on translation in plants during stress.

Most of these factors, or factor subunits, are multidomain proteins, as there are two isoforms of eIF4F in plants. eIF4F, which consists of eIF4E and eIF4G, and the eIFiso4F isoform, which includes eIFiso4E and eIFiso4G [14]. The eukaryotic translation initiation factor eIF4F is an essential component of the translation mechanism and also plays a role in developmental processes and stress relief in plants and animals. Both eIF4F isoforms are associated with resistance to various stresses. Mutations of both eIFIso4G have a negative effect on plant life by reducing the rate of translation. Cho et al. [15] found that eIF4F supports translation of mRNAs that have a structured 5′-UTR and lack a cap or contain multiple cistrons, whereas eIFiso4F prefers unstructured mRNAs or mRNAs with simple structures, which is also a feature of mRNAs translated during hypoxia. The study showed that eIFiso4G1 is required for hypoxia tolerance and SnRK1-mediated translation of specific mRNAs, including known hypoxia-responsive genes. This suggests that to respond rapidly to stress during hypoxia, SnRK1 phosphorylates eIFiso4G1, which increases protein biosynthesis and supports energy production, enabling stress adaptation.

Translation initiation factors respond to increases in temperature, so in plants, heat shock causes inhibition of cap-dependent translation; indeed, severe heat stress can completely inhibit this mechanism of translation. This suggests that cap-related initiation factors, such as eIF-4B or eIF-4F, may be altered after heat exposure. However, as mentioned above, plant translation factors differ significantly from those of animals and yeast in that plants contain not only eIF-4F but also the eIF-iso4F isoform. Gallie et al. [16] reported that while a number of translation factors (eIF-4E, eIF-4B and eIF-2α) are modified after heat shock in mammals, only eIF-4A and eIF-4B are thermally modified in wheat. This is mainly due to the tethered nature of plants and their adaptation to environmental changes to minimise the detrimental effects of stress [17]. One of the key steps in the initiation of translation is the establishment of the link between eIF4E and eIF4G. In most eukaryotes, this process is regulated by proteins that bind to eIF4E; in plants, CERES was discovered, which is an eIF4E-interacting protein [18]. Toribio and colleagues proposed that CERES, by reducing polysome load at ZT5 (zeitgeber time 5 (ZT5, 5 h after first light)), modulates glucose management in plants. CERES orthologues (homologues that also contain the canonical 4E-BS) were also analysed, and no analogues were identified in other eukaryotes outside the plant kingdom. It was concluded that unlike most eukaryotes, where the known translation regulators that bind to eIF4E are negative regulators of translation, CERES acts as a plant-specific translation initiator that enhances overall translation and regulates the translation of mRNA subsets at specific stages.

The group of eIF1 factors is involved in the stimulation and assembly of the 43S PIC (preinitiation complex) and includes eIF1 and eIF1A, both of which are highly conserved in all eukaryotes. The eIF1A factor acts as an mRNA scanner and AUG initiation codon locator. There are a number of papers dealing with the relationship between the eIF1A factor and salt tolerance in plants [19,20]. It has also been reported that eIF1A can regulate the expression of stress-related genes, including TOBLTP, GST, MnSOD, NtMPK1, poxN9 and CDPK1. The eukaryotic translation factor eIF5B in plants is of rather ancient origin and is a structural homologue of eubacterial IF2 [21]. Initiation factor 5B is a GTPase that acts late in translation initiation. Zhang et al. found [22] that the heat-sensitive mutant of *A. thaliana* (hot3-1) carries a mutation in the eIF5B1 gene. This mutant behaves like the wild type in the absence of stress, but cannot acclimate to high temperatures. There are three other known eIF5B genes in *A. thaliana*, but they do not substitute for eIF5B1. Polysome profiling and transcriptome analysis of hot3-1 plants show a delayed recovery of polysomes after heat stress and a decrease in the efficacy of proteins that protect against stress factors, demonstrating the important role of translational control in the early stages of heat acclimation.

eIF3 plays a key role in translation initiation by interacting with numerous factors as well as the ribosome. In plants, it is represented by a complex of 13 subunits [23]. The main role of eIF3 is to bind to the 40S subunit to participate in the formation of the 43S pre-initiation complex (PIC). At present, there are no precise data describing the structural composition of eIF3 in plants. Work on the effect of eIF3 overexpression on plant resistance to drought and other abiotic stresses has been published [24,25]. The eukaryotic translation initiation factor 4A (eIF4A) in the cell cytosol is present in different isoforms of eIF4A, namely eIF4A1, eIF4A2 and eIF4A3, and their expression is tightly regulated in cap-dependent translation. eIF4A is the prototype of a large family of RNA helicases; however, it has a minimal helicase core and lacks additional domains found in other helicases [26].

DEAD box helicases are known to play an important role in abiotic stress in flowering plants [27]. The role of eIF4A in plant tolerance to salinity and cold stress has been investigated and novel engineering pathways to maximise yield under suboptimal conditions have been proposed [28]. It is known that eIF4A is phosphorylated in response to hypoxia and heat shock [16,29].

### 2.2. RNA Binding Proteins

Post-transcriptional control of gene expression is achieved by regulating RNA metabolism (stability, splicing, polyadenylation and transport). RNA binding proteins (RBPs) play an important role in this process. They can make modifications directly or recruit other regulatory factors. The *A. thaliana* AtRBP45b protein contains three RNA recognition domains (RRMs) and has a Glu-rich region at the C-terminus that is involved in establishing protein–protein interactions. R. Mahalingam and colleagues showed that AtRBP45b interacts with the EDC4 protein (enhancer of decapping protein), which is involved in translational repression [30]. EDC4 interacts with decapping enzymes, which leads to translation repression and promotes RNA degradation. Other studies have shown that EDC4 interacts with RNA but does not have RRM motifs. It is possible that AtRBP45b is the platform on which EDC4 interacts with RNA. EDC4 is known to be localised in the cytoplasm in P-bodies, whose formation is associated with stress. In these areas, mRNA sequestration and translation repression occur. The researchers obtained results showing that AtRBP45b interacts with EDC4 as well as with the cap-binding protein CBP20 and the heat shock protein HSP40. The study used co-immunoprecipitation methods, two-hybrid analysis and FRET analysis (Förster resonance method). However, it is worth noting that the subject used was not a wild-type plant, but an AtRBP45b mutant line with restored expression. Another example of the involvement of RBPs in the regulation of stress responses is AtRBP-DR1, which is a positive regulator of the salicylate-mediated immune response in plants. It is possible that the effect of AtRBP-DR1 is mediated by an increase in the stability of the transcript of the SID2 gene (isochorismate synthase), which is involved in the synthesis of salicylic acid [31].

Genes encoding PR (pathogenesis-related) proteins are represented by 17 families, including a family containing the CAP (cysteine-rich secretory protein, antigen 5 and pathogenesis-related-1) domain. Aydin Akbudak et al. showed that members of this family are up-regulated in tomato in response to drought stress [32]. According to Chien et al. [33] in *A. thaliana*, short peptides (CARE peptides) obtained by cleavage of PR proteins containing the CAP domain (PROAtCAPE) play a regulatory role in response to different types of abiotic stress. Thus, in roots under salinity, the level of expression of a number of transcripts (PROAtCAPE1, PROAtCAPE3 and PROAtCAPE4) decreased under salinity, the content of the transcript of the PROAtCAPE3 and PROAtCAPE6 genes also decreased under drought and cold stress, whereas the expression of the PROAtCAPE7 transcript increased under cold stress. The level of the PROAtCAPE1 transcript was mainly regulated by salt, while the transcripts of the remaining PROAtCAPEs were regulated by more than two abiotic stresses. Under normal conditions, CAPE1 (a cleavage product of the PROAtCAPE1 gene) represses the expression of genes responsible for plant tolerance to salt. According to the authors of the article, CARE1 reduces the expression of genes involved in the synthesis of osmolytes (P5CS1 and GolS2). The level of endogenous CAPE1 in shoots was 25% of the corresponding level in roots. Since the expression of PROAtCAPE1 in shoots was not determined, the CARE1 found in shoots may be of root origin. Alternative splicing (AS), which generates different transcript variants, is a mechanism for the formation of resistance to abiotic stresses. AS is carried out with the participation of the spliceosome, a complex of nuclear ribonucleoproteins (snRNPs). Serine/arginine-rich proteins (SR proteins) play an important role in splice site recognition; they are associated with the spliceosome, but snRNPs are not [34]. These proteins are mainly involved in pre-mRNA splicing, but may also have other functions such as mRNA export, stability control and some others. They are characterised by the presence of one or two N-terminal RNA recognition motifs (RRMs) and a C-terminal arginine- or serine-containing domain that can be phosphorylated. The phosphorylation status controls the subcellular distribution. The role of SR proteins in plant responses to abiotic stress is important [35]. For example, the loss of function of the SR34b gene, whose up-regulation is induced by cadmium (Cd), results in increased accumulation of this metal ion and associated increased cytotoxicity. Importantly, the IRT1 gene, which encodes the putative Cd transporter, is absent in the sr34b mutant, suggesting the presence of mechanisms to control Cd tolerance using the SR protein [36]. In another study, *A. thaliana* RS40 and RS41 were found to interact with the HOS5 protein (osmotic stress response protein). RS40 and RS41 knockout mutants showed hypersensitivity to salt and ABA, as well as intron retention in many stress-related genes [37].

Six gene families (small heat shock proteins) have been identified based on localisation and sequence similarity. The expression of these proteins is induced by exposure to high temperatures and other types of abiotic stress. The accumulation of these proteins in response to adverse conditions correlates with stress resistance. For example, in the study of chloroplast sHsps in *Agrostis stolonifera*, several variants of these proteins were isolated: sHsp26.2 in the heat-resistant variant of the plant and identical sHsp26.2m in the heat-sensitive variant, which contained a point mutation producing a premature stop codon [38]. Another example is the presence of additional Hsp25 polypeptides in heat-resistant variants of *Agrostis palustris* (cited in [39]). It is known that sHsps act as molecular chaperones by binding to unstable conformations of various stress-forming proteins. sHsps, together with other classes of chaperones, ensure the formation of the correct conformation and facilitate folding. They can also promote trafficking to a specific subcellular compartment or participate in the degradation of misfolded proteins.

### 2.3. 5′-Untranslated Regions’ Unstructured Features

The 5′ untranslated regions are extremely important for both the canonical and alternative translation processes in both prokaryotes and eukaryotes. Plants are no exception. The 5′-UTR is involved in translation control in two separate but related ways: (1) it acts as a container for specific regulatory elements [40], and (2) it acts on its own, even without distinguishable patterns, due to its structure and nucleotide composition. If all is clear with the first type of determination, we will look in detail at various regulatory elements and patterns localised in the 5′-UTR. However, there is no specific consensus on the role of the 5′-UTR per se. In this section, we attempt to discuss what is currently known about the role of the 5′-UTR in shaping the response to stress at the translational level.

It is known that heat stress represses the translation of almost all genes. However, this is not the case for heat shock proteins, whose translation, contrary to expectations, is increased. The 5′-UTR is one of the factors that determines translation under conditions of a sudden increase in temperature. For the 5′-UTR of the Hsp81-3 protein gene, it was shown [41] that this untranslated region provides a scanning mechanism for translation initiation (to prove this, the authors integrated uAUG) for both capped and non-capped mRNAs; the level of translation of reporter genes under the control of this 5′-UTR was also shown to be indifferent to temperature changes. Unfortunately, the authors do not provide a detailed mechanism for the described process.

A likely answer to the above question is a recent study [42] whose authors found (in HeLa cell culture) that $N^{6}$-methyladenosine ($m^{6}A$) in the proximal part of the 5′-UTR is able to directly bind eIF3, which in turn recruits the 43S complex. In fact, the initiation mechanism presented is an alternative to both Cap-dependent and IRES-dependent translation initiation mechanisms Figure 3. In this work, it was found that this initiation mechanism is activated in response to stress factors, in particular heat stress, when the Cap-dependent translation mechanism is suppressed by inhibition of the eIF4E factor. Although the authors of this study demonstrated the pattern they discovered in HeLa cell culture, there is no reason to believe that such a mechanism would not work in plants. For *A. thaliana*, mRNA isoforms are shown that differ in the 5′-UTR and are characteristic of different temperature conditions. According to the authors of study [11], this may be due to the different stability of the 5′-UTR of such transcripts. A striking example is the expression of the gene encoding the PSY protein (phytoene synthase), which is involved in carotenoid metabolism. Two isoforms of the 5′ UTR have been identified in the mRNA of this protein: a long and a short one. The long one contains a stem-loop structure which, according to the researchers, could be a riboswitch that reacts to carotenoids or their metabolites. The short one does not contain such a secondary structure and can therefore rapidly increase the PSY level, which is necessary for the synthesis of abscisic acid, for example, under salinity [43].

The context of the proximal parts of the 5′-UTR may also affect the efficiency of translation under conditions of heat stress [44] and low temperature [45]. Although it can be difficult to isolate specific regulatory regions, the intrinsic characteristics of the 5′-UTR are length and GC content [46]. The authors of this study showed that the rice (*Oryza sativa*) translate (i.e., the predominant fraction of translated mRNAs) can be characterised by the predominance of short and GC-rich 5′-UTR sequences. In our opinion, it is necessary to consider in detail the role of the length and nucleotide composition of the 5′-UTR in plants under the action of stress factors and in the absence of stress. At the same time, particular attention should be paid to the development of methods that will allow the contribution of these factors (length and composition) to be assessed and those that have not yet been identified to be excluded, such as regions with a developed secondary and tertiary structure. Going back in time, we also find it extremely interesting to be able to demonstrate the presence in plants of an alternative scanning mechanism for translation initiation involving $N^{6}$-methyladenosine.

### 2.4. IRES

As previously stated, translation initiation is mainly triggered by the 5′-cap. However, alternative translation initiation mechanisms are activated in some instances, such as in the case of viral RNA [47], apoptosis [48] or mitosis [49,50]. Here, the small ribosomal subunit binds to the internal ribosome entry site (IRES), found in the 5′-UTR. The process of initiation relies on specific protein molecules, known as IRES Trans-Acting Factors (ITAFs), to mediate the process [51]. The method of IRES-driven translation initiation was initially outlined for picornavirus RNA [52]. Heat stress is a condition where cap-dependent translation is considerably hindered, but HSP-encoding mRNA translation functions optimally. There is evidence to suggest that stress triggers the intrinsic initiation of gene translation (HSP) [53]. Additionally, certain maize germ mRNAs are able to initiate cap-independent translation during germination [54].

Recently, the link between translation initiation via IRES and G-quadruplexes has been partly clarified [55] (this is detailed in the G-quadruplexes section). The homeodomain transcription factor WUSCHEL (WUS) protein preserves a pool of totipotent cells in the *A. thaliana* shoot apical meristem. The expression of this gene is managed by the IRES and the AtLa1 protein, which is an RNA-binding factor. It has been shown that uncharacterised cells in *A. thaliana*, in response to harmful external factors, initiate IRES-based translation of WUS mRNA under the supervision of the AtLa1 protein [56].

Despite the lack of data on the involvement of IRESes in stress regulation in plants, there are instances of using these regulatory elements to develop transgenic plants that are resistant to salinity, for instance [57,58]. Considering the future of research in this area, it is likely that researchers will concentrate on genome-wide analysis of the distribution of IRES and their functional annotation, specifically in relation to the response to abiotic stress.

### 2.5. Leaderless mRNAs

Leaderless mRNAs, i.e., mRNAs without a 5′-UTR, are of particular interest. Originally discovered in eubacteria [59,60], they were later found in archaea and eukaryotes. Given that the main mechanism of translation initiation in both prokaryotes and eukaryotes requires a 5′ UTR, it would seem that such a model would not work. However, it has been shown that in *E. coli*, for example, leaderless mRNAs gain an advantage in translation during cold stress (apparently due to the lack of secondary and tertiary structure developed prior to initiation of the AUG) [59]. Translation of leaderless mRNAs has also been reported in mammalian mitochondria (known to be of prokaryotic origin) [61,62]. It has been shown that only the translation initiation factors IF-2mt and IF-3mt are required for this process; other factors are not necessary. Nuclear-derived leaderless mRNAs can bind to the 80S ribosome-Met-tRNA complex in mammalian cells in the absence of any translation initiation factors [63,64].

If we discuss the representatives of the plant kingdom (Viridiplantae), similar data were only obtained for mitochondria from *Chlamydomonas reinhardtii* [65]. In this organism, eight protein-coding genes are transcribed as two polycistronic transcripts, which are subsequently hydrolysed to form transcripts that begin with a start codon and have an extended 3′ UTR. The authors of this study showed that such mRNA molecules are looped, forming a kind of pseudo-5′-UTR. The study of leaderless RNAs requires further large-scale experiments and raises the following open questions: Are the patterns found in mammalian and bacterial cells applicable to plants? How common are leaderless mRNAs in higher plants? Do they have an advantage in transcription under stressful conditions? What is the rate of degradation of leaderless mRNAs? Are there mechanisms that protect leaderless mRNAs from rapid degradation?

### 2.6. uORFs

Upstream open reading frames (uORFs) located in the 5′ untranslated region (5′-UTR) of some plant mRNAs can regulate translation of the main open reading frame (mORF) encoding the major protein product of the transcript [66,67,68,69]. uORFs regulate translation of the underlying core ORF through small protein signalling molecules. In plants, uORFs have been found in 24–30% of the 5′-UTR region of mRNA [40,70]. The uORF prevalence approximation is based on the use of AUG as the start codon, but ribosome profiling studies show that non-canonical codons (e.g., ACG) in uORF can also serve as start codons [71].

In response to abiotic stresses such as drought, salt stress or low temperature, gene expression in plants can change. uORFs can play an important role in this process by controlling the translation of mORFs (main ORFs). Typically, uORFs inhibit translation initiation at the underlying CDS coding sequence. The inhibitory mechanism of uORFs is explained by the fact that in eukaryotes, the 43S preinitiation complex binds to the 5′-cap structure of the mRNA, processively scans in the 5′-3′ direction and initiates translation at the first initiation codon encountered, which is in the optimal environment [72]. The 43S pre-initiation complex consists of several factors, including eukaryotic initiation factor (eIF)-3, eIF1, eIF1A, the eIF2/GTP/Met-tRNAiMet ternary complex and the small 40S ribosomal subunit [73]. Dissociation of the eIF2 ternary complex and other initiation factors during uORF translation is thought to cause repression of subsequent translational reinitiation in downstream CDSs [74]. uORF-mediated inhibition of ORF translation can occur either passively, in the case of ribosome dissociation following uORF translation, or actively, in the case of ribosome arrest caused by uORF translation [68,69]. The arrest may isolate translating ribosomes, blocking their access to downstream ORFs, or be detected as abnormal translation termination, triggering transcript destruction via the nonsense-mediated decay (NMD) mRNA mechanism [75,76]. In each case, translation of the downstream mORF is inhibited.

Using a variety of approaches, a significant number of genes with uORF-encoded peptides have been identified that are highly conserved among different representatives of eukaryotes, including plants. These uORFs have been independently named upstream conserved coding regions (uCCs) or conserved peptide uORFs (CPuORFs). Relatively few of the plant CPuORFs have a known biological function; those that have been characterised are involved in the translational regulation of mORFs in a largely metabolite-dependent manner. CPuORFs play a role in regulating transcription factors such as bZIP (basic leucine zipper) and bHLH (basic helix–loop–helix) that are induced in response to abiotic stresses. The most important and best studied example of metabolite-controlled conserved peptide-dependent regulation of uORF translation in plants is the bZIP transcription factors of the *A. thaliana* C/S1 group (bZIP1, bZIP2/GBP5, ATB2/bZIP11, bZIP44 and bZIP53). S1-bZIP plays an important role in plant adaptation to adverse conditions [77,78]. It has been confirmed that S1-bZIP plays an important role in plant response to abiotic stresses such as low temperature [79,80], drought [81,82] and salinity [82]. It has been shown that the C-/S1-bZIP-SnRK1 complex is involved in the reprogramming of primary metabolism related to carbohydrates and amino acids and induces tolerance to salt stress through ABA-independent signalling in *A. thaliana* roots [82].

In plants, polyamines play an important role in adaptation to adverse conditions and stress resistance in general. Polyamines influence the adaptation of different plant species to different abiotic stresses such as drought [83,84], salt stress [85], high temperature and cold stress [86]. A key target of polyamine-regulated translational control in many eukaryotic systems is AdoMetDC (AMD1) [87]. The 5′ leader of the plant AdoMetDC mRNA contains two conserved overlapping uORFs, one small and one large. The small uORF consists of only three codons. The small uORF is in the range of 40–65 codons and shows greater conservation at the amino acid level. The C-terminus of a small uORF always ends with PS [88]. Overexpression of AdoMetDC mRNA using a construct in which uORFs are eliminated by mutation of the start codons results in increased enzymatic activity of AdoMetDC, disruption of polyamine homeostasis, and abnormal plant growth and development [89].

### 2.7. RNA Thermometers

RNA thermometers are sections of mRNA that can change their secondary and tertiary structure when the temperature changes in the physiological range. These regulatory elements have been most extensively studied in prokaryotes. In bacteria, in particular, the fundamental possibility of using almost any biological molecule as a temperature sensor has been demonstrated: DNA, RNA, proteins and lipids [90,91]. Bacterial RNA thermometers perform a wide range of functions, including induction of genes responsible for invasion (in the case of warm-blooded pathogens); quorum sensing; adaptation to temperature increase; adaptation to lower temperatures; formation of biofilms; transition from lysogenic to lytic cycle in bacteriophage λ [92].

In animals, the situation with the knowledge of RNA thermometers is much more modest [93]. Among the regulatory elements that directly act as thermosensors, we can mention the 5′-UTR mRNA encoding the heat shock protein 90 (HSP90) in *Drosophila*, as well as the non-coding RNA Hsr1 (which is discussed below). In addition, the dependence of the intensity and site-specificity of RNA editing on temperature in *Drosophila* has been demonstrated [93].

The study of plant RNA thermometers is at an early stage. Reviews [94,95] demonstrate an approach based on the suggestion of similarity of molecular mechanisms in plants, prokaryotes and non-plant eukaryotes (mammals, protists and insects). One of the first RNA thermometers discovered in plants is the hairpin in the 5′ UTR of the PIF7 transcription factor. PIF7 is involved in the activation of genes responsible for the response to heat stress in *A. thaliana*. PIF7 also directly interacts with PIF4 (PHYTOCHROME INTERACTING FACTOR 4), which is involved in the control of auxin synthesis. A similar hairpin has also been described for the transcript encoding the transcription factor WRKY22 and for HSFA2, which in turn controls the expression of the heat shock proteins [96]. Hsr1 is a short, conserved RNA that controls the expression of heat shock proteins. Papers [93,97] found it in a wide range of organisms. At present, it seems to be a consequence of horizontal gene transfer or contamination. The Hsr1 gene has been shown to have a sequence identical to the bacterial voltage-gated chloride channel protein.

At first sight, two different mechanisms underlie the function of RNA thermometers in prokaryotes and eukaryotes. In eukaryotes, hairpins prevent the implementation of the scanning mechanism of translation initiation; the hairpin mechanically prevents the movement of the complex of the small subunit of the ribosome and the translation initiation factors from the 5′ cap to the start codon surrounded by the Kozak consensus sequence. In an experiment to study the structure of rice transcripts, only a few candidate sequences were found to play the role of RNA thermometers, and none of them melted at 42 °C [98].

Information on the role of RNA thermometers located in the 3′-untranslated region in regulating translation is extremely scarce. For Leishmania, the presence of a structured region in the 3′-UTR has been demonstrated, which undergoes partial melting with increasing temperature in the range of 26–37 °C. This may facilitate the cyclization of the mRNA molecule and the subsequent reinitiation of translation at elevated temperatures [94].

At present, the topic of RNA thermometers in plants raises more questions than it answers. In particular, the question of whether RNA thermometers are present in higher-plant mRNAs is of great interest. If the answer to this question is negative, then we believe it is necessary to clarify the mechanisms that prevent such a scheme of translation regulation. If the answer is yes, then what are the algorithms for finding such patterns? Can there be specific RNA sensors that are sensitive to temperature drops? In this situation, there are two possibilities: first, such thermometers change their structure with decreasing temperature to one that is more passable for ribosomes but with higher potential energy (for example, two low-molecular-weight hairpins instead of one high-molecular-weight hairpin, similar to the tryptophan operator); or as mRNA structures that block the expression of genes that themselves act as suppressors when the temperature is lowered. It remains to be seen whether plant RNA thermometers are integrated into larger regulatory regions (e.g., IRES). Could RNA thermometers be involved in the process of RNA interference?

### 2.8. Plant Ribosomes Also Act as an Independent Sensor

The authors of a recent study have shown that a decrease in the rate of translation under the influence of cold (and cycloheximide) leads to an increase in the concentration of calcium ions in the cytoplasm and nucleus. An increase in the concentration of Ca^2+^ in turn induces the expression of the CAMTA complex, which already triggers the expression of the CBF gene responsible for overcoming cold shock. The involvement of ribosomal proteins in this signalling pathway has been demonstrated, suggesting that ribosomes may act as independent thermosensors. Ribosome inactivating proteins are able to release adenine from RNA and DNA molecules [99]. Regarding models of *Hura crepitans*, *Phytolaeca americana* [100] and *Mesembryanthemum crystallinum*, ref. [101] showed that the levels of these proteins increase in ageing leaves and in leaves subjected to heat or osmotic stress. Changes in the population of ribosomal proteins during acclimation after cold stress (from Day 1 to 7) were shown in a *A. thaliana* model using mutants for the REIL (Rei-like) protein gene [102].

### 2.9. G-Quadruplexes

G-quadruplexes are secondary and tertiary structures within mRNA (and DNA) based on non-Watson–Crick interactions between consecutive guanines; Figure 4. This area of research is currently developing very dynamically [103]. Initially, G-quadruplexes were considered to be regulatory elements active at the transcriptional stage. However, with time, data are accumulating that G-quadruplexes also play an important role in the regulation of translation. In addition, like many regulatory elements, GQS (G-quadruplex sequence) modulates translation in two ways. Typically, GQS suppresses ribosome targeting of the 5′-UTR. Alternatively, the GQS of various regulatory elements with a developed secondary and tertiary structure (IRES, riboswitches, etc.) can also be stabilised. GQs can also serve as binding sites for various proteins and small molecules. Such proteins can either stabilise (fragile X mental retardation protein (FMRP)) the structure of G-quadruplexes (translation repression), act as specific helicases (RHAU/DHX36), or increase the intensity of translation by reducing the complexity of the mRNA structure [104]. An example [105] for plants is the regulation of the gene encoding the SUPPRESSOR OF MAX2 1-LIKE4/5 (SMXL4/5) protein. This protein is responsible for the development of the phloem in vascular plants. Using *Populus tremula* and *A. thaliana* as examples, the authors of this study showed that SMXL4/5 contains a 5′-UTR sequence capable of forming GQS. Another zinc finger protein, JULGI (JUL), binds to this region as part of the 5′-UTR and induces the formation of a G-quadruplex (i.e., in the native version, without JUL, it is not formed), which inhibits translation of SMXL4/5. In silico analysis (using BLASTp and FIMO) identified over 400 proteins potentially capable of binding to GQS [106].

The interaction of small molecules and G-quadruplexes is similar to that of proteins. Some compounds are able to stabilise the structure of GQSs, while others cause the breakdown of these mRNA structures and increase their permeability to ribosomes. In this case, G-quadruplexes can act as riboswitches. In addition, synthetic aptamers can be constructed based on sequences containing GQS [107]. The stability of GQSs is also determined by their dehydration, i.e., their increase can be facilitated by increasing the concentration of ions or osmolytes in the cytosol. An equally intriguing hypothesis was put forward by the authors of study [108]. In their work, they demonstrated the ability of ultraviolet radiation to destabilise GQS under in vitro conditions Figure 4. If such a mechanism occurs in living plant cells, it may act as a potential sensor responsible for the response to increasing solar radiation and related stress conditions (drought and temperature stress).

GQS have also been implicated in liquid–liquid phase separation in *A. thaliana*. SHORT ROOT (SHR) [109]. An active involvement of SHR mRNA in liquid–liquid phase separation has been demonstrated. The mRNA of this gene was also shown to form a G-quadruplex under certain conditions (with a change in the concentration of potassium ions). Using normal and mutant variants of the mRNA sequence, the authors of this paper demonstrated the involvement of GQS in the process of phase separation. Given that phase separation can occur during stress responses (e.g., in the form of stress granule formation), this study opens up another avenue of research—it is unlikely that this G-quadruplex alone is involved in phase separation. A new stage in the study of G-quadruplexes has been the use of genome-wide and whole-transcriptome approaches, based on sequencing of all transcripts and subsequent screening of potential GQSs (and sometimes verification of individual candidate sequences). Complete translational analysis revealed the putative involvement of G-qualruplexes in the regulation of translation under heat stress in *A. thaliana* [110].

Genome-wide analysis (on the *A. thaliana* model) using Markov models in combination with GO enrichment analysis showed significant changes in some gene groups. Overrepresented: catalytic activity, protein amino acid phosphorylation, transferase activity, etc. Under-represented: translation, gene expression, RNA binding, etc. [111]. Potential GQSs associated with responses to various types of abiotic stress (including hypoxia and nutrient deficiency) have also been found in the maize genome. Quite curious is the presence of GQSs in the antisense strand in the 5′ UTR and at the 5′ end of some genes responsible for acclimation under these stress conditions (a similar pattern is also characteristic of the wheat genome [112]). However, GQSs that directly regulate broadcasting have not been identified [113]. A genome-wide search for potential GQSs using barley as an example showed an increased representation of these motifs within 1Kb of the start codons of genes (300 per 1e6 versus 100 per 1e6 for the entire genome) [114]. These data are also confirmed by experimental work carried out on the wheat genome [112].

The widespread use of in silico methods, as seen in the above work, can be explained by the availability of these methods for direct use. Nevertheless, we believe that there is a need to develop evidence-based methods using molecular cloning tools and whole-genome techniques to detect both GQSs directly in transcripts and protein–GQS interactions. An interesting approach has been proposed in study [115]. It is based on blocking the GQS-containing region for the RNA-dependent DNA polymerase. Thus, using specific primers for reverse transcription, it is possible to detect the presence of a formed G-quadruplex in mRNA. The modulation of the GQ structure is based on the selectivity of this motif for potassium ions Figure 4, i.e., a G-quadruplex is formed in the presence of K^+^ but not in the presence of Li^+^ [115]. This approach has been further developed and improved for full transcriptome searches and GQS analysis [116,117]. There is also a method based on the precipitation of RNA containing GQS and further sequencing of these molecules [118].

Considering that most studies on the contribution of G-quadruplexes to the regulation of translation have been carried out in animals, it is reasonable to consider the role of GQS in plants by analogy. However, additional avenues for future research can be identified: (i) the study of GQS-binding proteins in plants, (ii) the distribution of GQS among differentially expressed genes under stress provoking the accumulation of osmolytes (cold, drought), (iii) the prospects of using GQS as riboswitches in plants, in particular as aptamers for specific low-molecular-weight ligands and as RNA thermometers (i.e., the stability dynamics of these elements under *in vivo* conditions at different temperatures is of interest).

### 2.10. Kozak Consensus Sequence

One of the key steps in the control of gene expression (as mentioned above) is the mechanism of regulation of translation initiation, in which the sequence context around the AUG start codon, which forms the translation initiation site (TIS), plays an important role. To date, the main model of translation initiation is considered to be that described by Marilyn Kozak, according to which the 40S ribosomal subunit associated with the 5′-terminal region of the mRNA, together with a number of translation initiation factors, begins a linear scan along the mRNA until it reaches the start codon, most commonly AUG [119]. It is generally accepted that successful recognition of the AUG start codon by eukaryotic ribosomes depends on its nucleotide environment. Thus, in the 1980s, Kozak carried out research in which 699 vertebrate mRNAs were analysed, resulting in the optimal TIS consensus sequence, known as the Kozak motif gccgcC(A/G)ccAUGG, where AUG represents the start codon. In this case, the position of the purine at −3 (97% of mRNA) and G (61%) at +4 (where A in AUG is at +1) forms the optimal sequence for TIS recognition and subsequent translation initiation [120].

In addition, sequences surrounding the start codon were classified based on the presence of two conserved nucleotides at positions −3 and +4. Thus, the Kozak motif NNN(A/G)NNAUGG (where N is any base), which contains both important nucleotides, is called “strong”, whereas the sequence GCC(A/G)CCAUGG is usually called “optimal”, while NNN(A/G)NNAUG(A/C/U) or NNN(C/U)NNAUGG is “adequate” because it contains only one of these nucleotides; “weak” is the sequence NNN(C/U)NNAUG(A/C/U), which does not contain any of the key nucleotides [121].

Further studies of other taxonomic groups, including plants, extended the understanding of the consensus sequence around TIS [122]; however, significant differences between taxa were observed, which contradicted the initial belief in the universality of the consensus Kozak for all eukaryotes. Studies by Gupta et al. presented a comparative genome-wide analysis of 14 plant species (seven monocots and seven dicots), taking 500 genes with the highest protein content and 500 genes with the lowest protein content. The results showed a direct relationship between highly conserved nucleotide positions at positions −3A/G and +4G and high protein levels in monocots and dicots. In addition, the C at position +5 also showed an increase in translation efficiency, in contrast to Kozak’s early data where the nucleotide at position +5 did not enhance translation initiation [123]. Comparative analysis also revealed a significant predisposition of the A/C nucleotides at position −2 to highly expressed genes in both monocots and dicots. Interestingly, the −2A/C position is found in the TIS consensus sequences of chordates, invertebrates, unicellular fungi and protists, making it rather conserved [124]. As a result, the sequences GCNAUGGC, AANAUGGC, and GCNAUGGC have been established as the archetypal TIS signals in monocots, dicots, and plants in general, respectively [125].

Regulation of gene expression at the translational level enables cells to mount rapid and reversible responses to sudden environmental changes. At the same time, the use of Kozak sequences capable of increasing the level of gene expression opens up a wide range of possibilities for practical application. For example, in studies aimed at increasing the resistance of *Oryza sativa* L. to dehydration stress, Zhou et al. produced transgenic rice plants expressing the fungal gene for glutamate dehydrogenase, which can act as a stress-responsive enzyme to detoxify high levels of intracellular ammonia and produce glutamate for proline synthesis under stress. To increase the level of translation of eukaryotic genes in transgenic rice plants, the Kozak sequence (GCCACC) was added before the MgGDH start codon. As the authors note, as a result of the high level of expression of the MgGDH gene, the transgenic rice plants showed an increase in tolerance to dehydration stress [126]. The data obtained are consistent with previous results showing a high level of expression of genes containing strong Kozak sequences [125].

However, there are studies showing that site-directed mutagenesis experiments with the Kozak sequence can tune the physiological response to stress. For example, a recent study showed that in response to stress, *Saccharomyces cerevisiae* produces ribosomes lacking certain proteins (for example, ΔRps26), which initiate the activation of certain signalling pathways (Hog1 and Rim101) that respond to osmotic and pH stress, respectively, making the cells more resistant to high-salt and high-pH conditions. At the same time, Rps26 preferentially initiates translation of mRNAs containing A at positions −2 and −4, whereas mRNAs containing G at positions −2/−4 are translated with ΔRps26 [127]. It is also noted that ΔRps26 does not activate the cell wall integrity (CWI) pathway because the Rho1 gene of the CWI pathway contains an A at position −2/−4. As a result of site-directed mutagenesis, a mutant Rho1 containing −2/−4G was obtained. At the same time, cells containing modified Rho1 showed increased resistance to zymolysis with depletion of ΔRps26, in contrast to cells containing native Rho1. This site-directed mutagenesis method was also applied to the Rnr2 and Ras2 genes of the DNA damage response and filamentation pathways, respectively. As before, the modified genes were found to be sensitive to the accumulation of ΔRps26 in the cell, in contrast to the native ones. Thus, by introducing point mutations in the Kozak sequence of individual genes, the authors were able to reprogramme the cellular response to severe salt stress, including the pathways of filamentation, activation of DNA repair and strengthening of the cell wall [128].

The relationship between the start codon environment and the regulation of translation under stress in plants is not obvious. At present, it is difficult to provide a clear answer as to whether this relationship can, in principle, exist. The environment of the start codon is rarely modified. Nevertheless, several main avenues can be outlined for further study of this issue: the first is an analysis of the representation of different types of Kozak consensus sequences at all stages of the realisation of genetic information, and the enrichment and bias in the frequencies of these types under stress. Second, and no less interesting, is the presence of alternative start codons and the extent to which they are used under stress. The third approach is to study epigenetic modifications of nucleotides in the vicinity of the start codon in plants.

### 2.11. Codon Usage

Codon bias is a well-established occurrence that involves the favouring of certain synonymous triplets over others in protein-coding sequences. It has been demonstrated [129] that optimal and conserved codons modify the rate of translation elongation under heat stress conditions in *A. thaliana* plants. Furthermore, an intriguing study revealed a preference for the most common codons in the translation of heat shock proteins in tomato [130]. Using a rice model, a study demonstrated the correlation between certain triplets preference and the plant’s adaptation and survival in drought conditions [131]. To achieve this, predictors such as effective number of codon usage (ENC), codon adaptation index (CAI) and relative synonymous codon usage (RSCU) were employed. A genome-wide analysis of codon usage in rice indicated that genes expressing at elevated levels under abiotic stress conditions are marked by a higher frequency of G/C occurrence at the final position in triplets. For *Liriodendron chinense*, study [132] exhibited the creation of a particular codon bias in genes of the gibberellin oxidase family via selection. It is intriguing that diverse plant species conduct the selection of a reliable information pathway formed by codon bias in various methods. For instance, in *A. thaliana* and soybean, Mitogen Activated Protein Kinase genes exhibit a high AT content, whereas in rice, the AT/GC ratio is well-balanced [133].

The association of this phenomenon with translation control is evident, owing to varying concentrations of transfer RNAs and frequencies of synonymous codons. Essentially, independent and non-competing transmission channels are established. We believe ambiguity in the examination of codon usage stems from identifying more formal patterns in codon preference schemes. The existence of stable translation channels formed by preferred codons is still not fully understood, and whether these channels have a capacity is uncertain.

### 2.12. lncRNA

Natural antisense transcripts NATs (natural antisense transcripts) are a type of long non-coding RNA. NATs are transcribed from the non-coding strand of protein-coding regions of the genome and are involved in the regulation of transcription and translation of the complementary (sense) strand product. In this case, they are called cis-NATs. When the target of the NAT is mRNA translated from another gene segment, such molecules are called trans-NATs. Julia Bailey-Serres et al. showed that some cis-NATs contain open reading frames encoding small peptides. Translation of these frames, called sORFs (small open reading frames), correlated with increased translation of the corresponding sense mRNAs. The authors suggest that translation of sORFs may stabilise or promote the targeting of cis-NATs. The authors carried out studies in *A. thaliana* under conditions of phosphorus deficiency and showed that the genes subject to the above regulation include two members of the ABC family of ATP-binding transporters (ABCG2 and ABCG20) and one member of the receptor kinase family (POLLEN-SPECIFIC RECEPTOR-LIKE KINASE 7). These genes are involved in the regulation of mineral uptake as well as in the formation of lateral roots, which is one of the conditions for adaptation to phosphorus deficiency [134].

A group of scientists led by Jinhui Chen, using third-generation sequencing (Iso-Seq) and high-throughput RNA sequencing, showed an increase in the transcript levels of five genes of eIF2Ds translation initiation factors, as well as a number of long non-coding RNAs, including lncPs3, against the background of heat stress in *Populus simonii* (Chinese poplar). Bioinformatic methods were used to predict the interaction of lncPs3 with eIF2D proteins. Thus, the authors suggest the involvement of long non-coding RNAs in the regulation of translation under heat stress in poplar [135].

An interesting study [136] demonstrated the involvement of an lncRNA called CRIR1 (cold-responsive intergenic lncRNA 1) in the response to cold stress in cassava (*Manihot esculenta* Crantz). Regulation by this lncRNA bypasses the regulation of C-repeat bindingin gene expression in several ways, including transcription factors (CBFs), but due to the close interaction of this lncRNA with MeCSP5 (a cold shock domain-containing protein), the link between this regulatory process and the translational process is not entirely clear.

### 2.13. MicroRNAs

MicroRNAs, small non-coding RNA molecules of 18–25 nucleotides in length, are involved in the transcriptional and translational regulation of gene expression through RNA interference. The formation of miRNAs can be roughly divided into three steps. The first step is the transcription of an approximately 1000 bp miRNA precursor by RNA polymerase II (or, in some cases, RNA polymerase III). In the second step, this precursor molecule is cleaved by RNase III DCL-1 (Dicer-Like 1), resulting in the formation of pre-miRNA, which is a miRNA:miRNA duplex. Pre-miRNA is exported from the nucleus to the cytoplasm with the help of the transport protein HASTY. At the third stage in the cytoplasm, the pre-miRNA binds to the Argonaute (AGO) protein, which is part of the RNA-induced silencing complex (RISC). Gene silencing can be achieved by degrading the mRNA or preventing its translation. If the miRNA is fully complementary to the target mRNA, AGO can cleave the mRNA and lead to its direct degradation. If there is no complete complementarity, shutdown is achieved by preventing translation [137]. Using an in vitro model, Iwakawa et al. showed that the AGO1-RISC complex can be involved in both target mRNA cleavage and translation repression. At the same time, several different mechanisms for translational repression have been reported, depending on the location of miRNA target sites. For example, when the AGO1-RISC complex is localised in the 3′-UTR region, translation is inhibited at the stage of 48S complex formation. This mechanism is similar to that in animal cells. The repression is thought to be mediated by eIF4E, eIF4G and eIF4A.When the binding site of the AGO1-RISC complex is located in the ORF or the 5′-UTR region, repression is achieved by sterically blocking the progress of the ribosome or its landing on the mRNA [138]. In general, the results of recent studies suggest that plant miRNAs have sites of complementarity to the 5′-UTRs of target transcripts, and most likely translational repression is mediated by AGO1-RISC binding to the 5′-regions, whereas in animals the preferred mechanism is 3′-mediated repression [139].

In plants, microRNAs are actively involved in the regulation of gene expression at the translational level under various types of abiotic stress. At the same time, the targets of known miRNAs are very often transcription factors or proteins involved in auxin signalling, resulting in a systemic effect. In some cases, the targets are proteins specifically involved in the regulation of a particular process (heat shock proteins under high-temperature stress; membrane ion transporters under salt stress). We now look at some of the best studied examples of translational regulation by miRNAs under abiotic stress in plants.

The predicted target of mir393 is the mRNA of the TIR1 protein. TIR1 is a component of the ubiquitin ligase complex whose activity is directed to Aux/IAA repressor proteins associated with promoters of auxin-regulated genes. Thus, an increased level of mir393 leads to a reduction in the amplitude of the auxin response and slows down plant growth under unfavourable conditions. The expression of this microRNA increases in response to cold stress and drought (cited in [140]).

In a group led by Youxin Jin, microRNAs of the miR169 family were found. They found that miR169g is induced by drought, while miR169n is induced by salt stress. Sequences were found in the promoters of these microRNA genes that bind transcription factors associated with drought and salinity, respectively, providing further evidence for their inducible expression. It was also shown that the target of miR169 is the transcription factor of the NF-Y family [141]. This transcription factor was previously shown to be involved in the response to drought stress (cited in [142]).

The miR395 family is an important regulator of sulfate assimilation in *A. thaliana*. The targets of these miRNAs are high-affinity sulfate transporters as well as three enzymes of the ATP sulfurylase class involved in the process of $\mathrm{SO}{}_{4}^{2-}$ uptake [143]. The level of miR395 varies depending on the exposure to different metabolites that regulate sulphate assimilation [144].

Chiou et al. showed that the regulation of phosphorus assimilation is controlled by miR399 through the downregulation of E2-ubiquitin ligase. The authors propose an elegant scheme for the regulation of phosphorus fluxes in wild-type and transgenic plants expressing miR399, which is not expressed at all in intact plants in the absence of phosphorus deficiency [145]. Recent studies report that miR399 has a so-called pseudo-target that can complementarily bind miRNAs but does not contain a cleavage site. It has been shown that inorganic phosphate deficiency in Arabidopsis increases the expression of IPS genes (induced by phosphate starvation), which, as pseudo-targets of miR399, regulate the expression of the true target of miR399, the E2-ubiquitin ligase (PHO2), involved in the renewal of the phosphate transporter [134].

Another microRNA involved in the response to mineral depletion is miR398, whose target is Cu-dependent superoxide desmutase (SOD). SOD is involved in the elimination of reactive oxygen species; therefore, under oxidative stress, the expression level of miR398 decreases, allowing the accumulation of SOD transcripts. However, as a metal-dependent enzyme, SOD is involved in the homeostasis of Cu^2+^ ions, and when this mineral nutrient is starved, the amount of miR398 increases [146].

Nitrogen availability affects the architecture of the root system through mir393, which has already been discussed in the context of drought. The target of this miRNA can be auxin receptors (TIR1, AFB1, AFB2 and AFB3), as well as transcription factor bHLH (basic helix–loop–helix) [147]. It is reported that mir393 is expressed under osmotic stress and, by cleaving its TIR1 and AFB2 targets, leads to the blocking of the auxin signal and lateral root initiation [148]. Another miRNA whose expression is controlled by nitrogen availability is miR167. Its target is one of the auxin receptors, ARF8. The presence of transport forms of nitrogen (glutamate) is a signal for the suppression of miR167 expression, which removes the block in the auxin signal and leads to the initiation of lateral root formation. miR167 expression is also downregulated under drought and high temperature, whereas miR167 levels increase under cold stress. A complex regulatory network involving miR167, mir393 and ARFs has been reported to regulate auxin-mediated signalling during osmotic stress (cited in [149]).

### 2.14. Global Changes in Plant mRNA Structure under Abiotic Stress

Global changes in mRNA structure are thematically closely related to the general characteristics of 5′-untranslated regions. Quite curious is the approach based on the complete transcriptomic analysis of double-stranded elements in the structure of transcripts [98]. The essence of the approach proposed by the authors—Structure-seq2—is to methylate in vivo adenine and cytosine as part of single-stranded mRNA regions. Such methylated bases cause the synthesis reaction to stop during sequencing. The researchers also took into account the fact that methylation of bases is not only avoided in the composition of double-stranded RNA regions, but also protected by proteins associated with nucleic acids. PIP-seq was used to address this issue. The study found that mRNAs are actively degraded under heat stress and that this process is closely linked to the melting of the secondary structure of these molecules. It was also found that there are no mechanisms for detecting an increase in temperature similar to those in prokaryotes (although this does not mean that they do not exist).

The analysis of the *A. thaliana* translatome under conditions of prolonged heat stress revealed the predominance of genes translated under these conditions whose mRNAs have a weakly structured or unstructured 5′-UTR, especially near the start codon [110]. The same study showed that a significant proportion of upregulated genes have consensus sequences in the 5′- and 3′-UTRs. A small number of papers in the declared area do not allow drawing of any far-reaching conclusions. It seems that systematic studies of global changes in mRNA structure are still lacking.

## 3. Methods for Studying Translation in Plants

The study of translation in plants is a rather complex process. First, as in the study of transcription, it is necessary to determine the abundance of genes in the composition of the transcriptome and translatome. Depending on the objectives of the study, it may be necessary to determine the secondary and tertiary structure of RNA, protein binding sites and the structure of certain regulatory elements already known in the transcriptome.

### 3.1. Analysis of the Involvement of mRNA in the Translation Process

Ribosomal profiling [150,151,152] makes it possible to identify a subset of polysome-bound mRNAs, as well as sites on messenger RNA molecules favoured by ribosomes (regions with rare codons, developed secondary structures, landing sites for specific proteins, etc.) [153]. Polysome profiling is an earlier and simpler version of ribosome profiling. This approach allows fractionation of polysomes according to the number of ribosomes in their composition. The disadvantage of this method is the lack of information about the ribosomal preferred sites. At the same time, high-resolution polysome profiling, both in terms of separation of polysome fractions and with sufficient sequencing depth, provides detailed information on the translation dynamics of specific transcripts. TRAP-seq [154,155] should be considered as an alternative to polysomal profiling, which does not reduce its effectiveness and also allows the analysis of changes in gene expression at the level of the entire translatome [156]. RNC-seq provides information on all translated transcripts, but does not provide quantitative information on the dynamics of translation itself or on specific nucleotide contexts.

As might be expected, the combination of these approaches increases both the overall resolution of the experiment and the range of information obtained. However, the main information provided by these methods is the transcript frequencies in the translation involvement gradient [157].

### 3.2. Examination of mRNA Secondary Structure

Given the variety of methods available to analyse the secondary structure of mRNA, only the advanced full transcriptome approaches are discussed here. A detailed overview of all modern approaches can be found in two reviews dedicated to the study of plant structure [118] and RNA structure in general [158].

Most modern methods of structural analysis are based on the identification of nucleotides that make up single-stranded regions of target molecules. Typically, the nucleotides in these regions are chemically modified or subjected to enzymatic degradation. It should be noted, however, that protection against chemical modification or enzymatic degradation can be provided both by binding nucleotides in secondary structures (hairpins, etc.) and by shielding protein molecules bound to mRNA. Another source of uncertainty is the conditions of the experiment itself: *in vivo* vs. *in vitro*. It is no secret that the conditions in a living cell can be very different from those in a test tube. In order to eliminate all these sources of uncertainty, a series of experiments were developed which, although not universal in themselves, together make it possible to understand the true state of the structure of mRNA in its natural environment. Modern methods for analysing the structure of RNA molecules are based on a number of general rules:For analysis under *in vivo* conditions, chemical modification of bases should be preferred since large molecules of RNases and proteases may have significant problems crossing the plasmalemma and, in the case of plants, the cell wall.Reagents used to modify unbound bases must not be toxic.-For example, dimethyl sulphate (DMS) [159], selectively methylating adenines and cytosines as part of single-stranded RNA regions; or N-cyclohexyl- N’-(2-morpholinoethyl)carbodiimide metho-p-toluenesulfonate (CMCT), which forms adducts with N1 and N3 of pseudouridine, N3 of uridine, and N1 of guanosine and inosine, as well as a combination of these two methods [160].-A separate method is icSHAPE [161], the essence of which is to modify free (single-stranded) RNA bases of all four types.-A similar approach is SHAPE-MaP [162].When analysing the temperature modulation of translation, one should take into account the increase in reactivity of modifying reagents and carry out the appropriate normalisation of the obtained data.To eliminate the ambiguity arising from the protection of bases by RNA-bound proteins, it is necessary to apply approaches based on selective combinatorial degradation:-double-stranded RNA,-single-stranded RNA,-proteins that protect the backbone of the RNA molecule.

The second group of methods for studying the structure comprises approaches based on cross-linking or ligation [158]. These techniques are based on the chemical combination of macromolecules that are closely spaced or actively interacting. The main example of this group of methods for analysing both the structure of RNA and RNA regions shielded by protein molecules is PIP-seq [163]. In fact, it is a further development of the previously known approach of RNase-mediated protein footprint sequencing [164]. In short, the sample is cross-linked (by formaldehyde treatment). Then, the sample is divided into several parts. Some of the samples are then deproteinised, while others are immediately processed with either duplex-specific or single-stranded RNA-specific nucleases. Deproteinised samples are also hydrolysed in the same way. Strand-specific library preparation and sequencing complete the analysis. The analysis of multiple fragments (after alignment, of course) gives an idea of the structure of the RNA and its associated proteins. The obvious disadvantage of this approach is that most of the manipulations are already carried out in in vitro conditions and not *in vivo*. However, this difficulty is easily overcome by combining PIP-seq with Structure-seq Figure 5 or similar methods, as was demonstrated in [98].

## 4. Concluding Remarks

We could not cover the entire process of translational regulation in plants under abiotic stress in a single review. For example, stop codon readthrough (SCR) has been demonstrated in many organisms. This phenomenon consists of the ribosome overriding the canonical stop codon. This mechanism is not well understood in plants. However, ref. [165] demonstrated SCR on the *A. thaliana* SCR model for 144 genes. Based on GO analysis, these genes can be divided into three main subgroups: translation, photosynthesis and response to abiotic stress. The peptides that are synthesised at the C-terminus during this process can be signals for localisation in peroxisomes, the nucleus or the membrane, as well as unstructured peptides without a clearly defined function. However, another interesting regulatory mechanism needs further experimental verification, for example, by analysing the stress response of plants. Another interesting study [166] showed an increase in the length of the poly-A tail of some Arabidopsis genes (including heat shock protein genes) in response to heat stress. However, the mechanism is not clear. The circadian rhythm also has a significant effect on the amount of translated mRNAs [167]. The specific mechanism remains unclear, and it is not entirely clear how to distinguish between the contributions of transcription and translation in this particular case. Many more similar examples could be presented. The study of translation in plants is the study of a complex system. As can be seen from the above, more than a dozen mechanisms may be involved in the control of translation. These processes vary in both speed and stability. We believe that this diversity allows, first, fine-tuning and rapid (from an evolutionary point of view) control of the translation of specific genes. Second, a large set of alternative disjoint possibilities organises multiple independent translation channels, which, in turn, expands the range of metabolic pathways for which co-orchestration becomes available. However, multithreading introduces additional difficulties in the analysis of data, especially those obtained from *ome-wide experiments. We believe that the next step in the development of the study of translation in general (and in plants in particular) is the creation of modern, updated databases that allow generalisation and prediction based on the data obtained. This is because, at the present stage, the analysis is already very difficult through the efforts of the researchers themselves. Finally, we would like to point out that the gaps in the model of translation in plants (including those under the influence of abiotic stress conditions, as an example of modifying factors) are rather an invitation to further work. If a mechanism is discovered in other taxa, it will be possible, by analogy, to demonstrate the existence of the same regulatory circuit for plants (since studies on this group of organisms are traditionally delayed). If there are difficulties with this approach, it will be no less intriguing to find out why this mechanism does not work in plants and the nature of these limitations.

## Figures and Tables

**Figure 1 cells-12-02445-f001:**
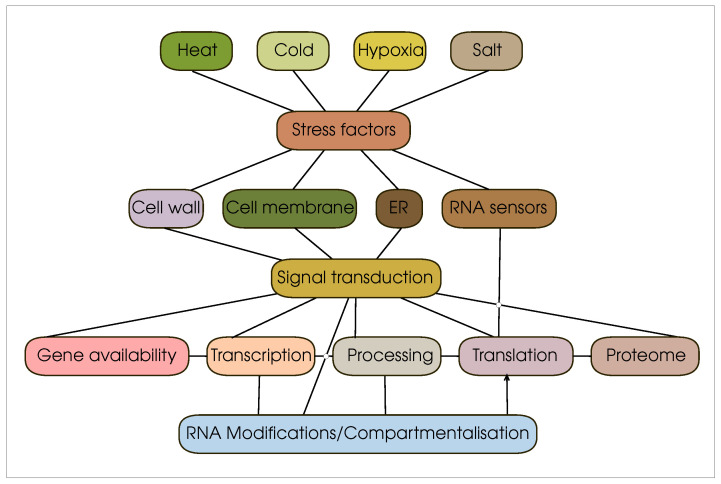
A general scheme of detection and coping with abiotic factors by plants. ER—endoplasmic reticulum.

**Figure 2 cells-12-02445-f002:**
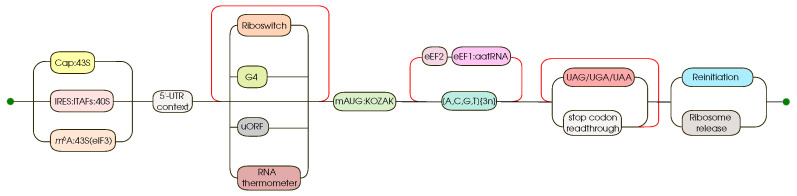
The journey of mRNA from translation initiation to the nascent peptide chain. Key regulatory elements and patterns: Cap—five-prime cap, 43S—43S preinitiation complex, IRES—internal ribosome entry site, ITAFs—IRES-transacting factors, $m^{6}A$—$N^{6}$-methyladenosine, G4—G-quadruplexes, uORF—upstream open reading frames, mAUG—main AUG, KOZAK—Kozak consensus sequences, aatRNA—aminoacetilated tRNA, eIF*—eukariotic initiation factors, eEF*—eukariotic elongation factors. Edges corresponding to the direct course of the translation process are marked in black; cycles within the translation are marked in red.

**Figure 3 cells-12-02445-f003:**
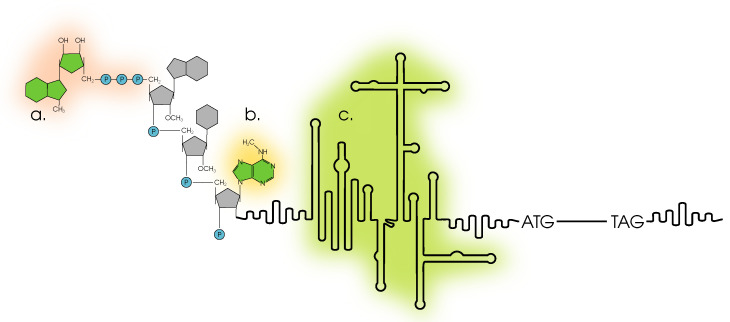
Main points of the translation initiation start. a. Five-prime cap (7-methylguanosine). b. $N^{6}$-methyladenosine ($m^{6}A$). c. IRES (internal ribosome entry site).

**Figure 4 cells-12-02445-f004:**
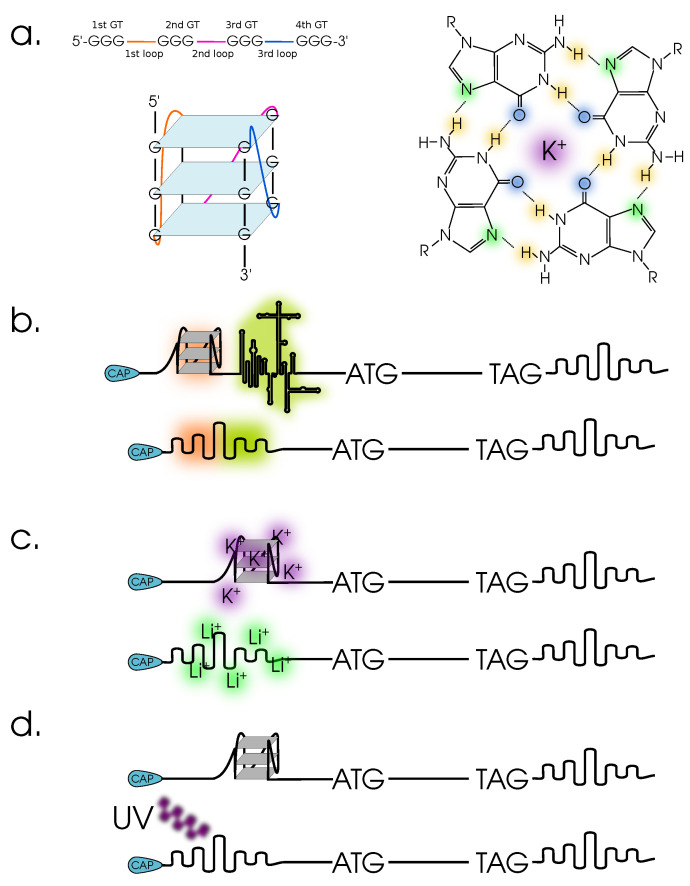
G-quadruplexes in plants. (**a**) Structure of the G-quadruplexes. (**b**) Participation of G-quadreplexes in stabilisation of secondary structure of mRNA (e.g., IRES). (**c**) Selectivity of G-quadruplexes to ions. (**d**) G-quadruplexes can act as UV sensors.

**Figure 5 cells-12-02445-f005:**
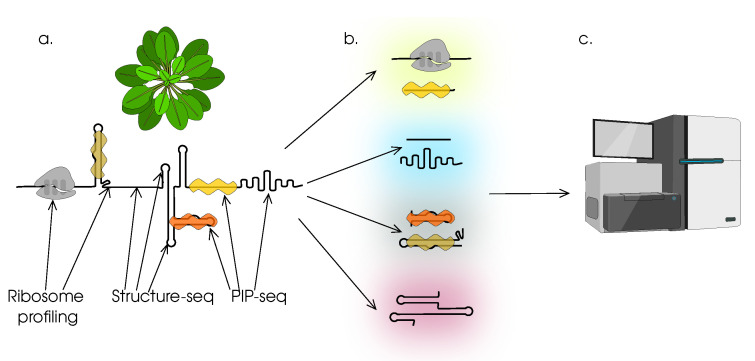
Methods of RNA structure investigation. (**a**) Approaches to RNA secondary structure and functional characterisation can cover the same areas of RNA. (**b**) Different types of mRNA organisation. (**c**) Short-read sequencing.

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
