# Peer review of "A Molecular Orchestration of Plant Translation under Abiotic Stress"

_cells, 2023, doi:10.3390/cells12202445_

Round 1
Reviewer 1 Report
The manuscript has been improved after the first revision round. The comments I've made in that first round have been taken into account, and changes brought accordingly.
I agree to publish this version of the manuscript.
Author Response
We thank the referee for the attention paid to our manuscript.
Reviewer 2 Report
The manuscripts is expected to present a review on regulation of plant translation under abiotic stress, as per the title. However, it seems that authors entirely fail to present the same rather included several aspects including RNA level regulation without any depth. Overall, the manuscript presents a random compilation of various aspects involved in abiotic stress regulation and that too has been very poorly written. Several sentences and even section headings do not make any sense.
Further there is no synthesis of text/discussion presenting the current status and future directions in the area of the manuscript.
Both figures included in the MS also seems vague with no relevance to the topic to the manuscript.
The writing of the MS is awkward at few places and sentences have not been framed correctly.
Author Response
We are grateful to the referee for reviewing our manuscript and providing constructive criticism. We have revised the language to be more clear and concise, and have included additional illustrations as suggested by the reviewers to enhance the manuscript's clarity and informativeness.
Reviewer 3 Report
The manuscript on 'Regulation of plant translation under abiotic stress. A molecular orchestration' is written well, and authors have very diligently covered all the topics with in the theme. My suggestion authors to include good illustrations of the mechanisms of plant translation control.
Author Response
We thank the referee for the attention paid to our manuscript. Following the recommendations of the reviewer, we added several new illustrations to the text of the manuscript, reflecting some features of the regulation of translation in plants.